# PSM: Learning Probabilistic Embeddings for Multi-scale Zero-shot Soundscape Mapping

## ABSTRACT

A soundscape is defined by the acoustic environment a person perceives at a location. In this work, we propose a framework for mapping soundscapes across the Earth. Since soundscapes involve sound distributions that span varying spatial scales, we represent locations with multi-scale satellite imagery and learn a joint representation among this imagery, audio, and text. To capture the inherent uncertainty in the soundscape of a location, we additionally design the representation space to be probabilistic. We also fuse ubiquitous metadata (including geolocation, time, and data source) to enable learning of spatially and temporally dynamic representations of soundscapes. We demonstrate the utility of our framework by creating large-scale soundscape maps integrating both audio and text with temporal control. To facilitate future research on this task, we also introduce a large-scale dataset, GeoSound, containing over $300k$ geotagged audio samples paired with both low- and high-resolution satellite imagery. We demonstrate that our method outperforms the existing state-of-the-art on both GeoSound and the existing SoundingEarth dataset. Our dataset and code will be made available at TBD.

## CCS CONCEPTS

• **Computing methodologies** → **Multimodal Learning**; **Self Supervised Learning**; **Remote Sensing**.

## KEYWORDS

Soundscape Mapping, Audio Visual Learning, Probabilistic Representation Learning

## 1 INTRODUCTION

Soundscape mapping involves understanding the relationship between locations on Earth and the distribution of sounds likely to be heard at those locations. The soundscape of an area is strongly correlated with psychological and physiological health [26]. Therefore, soundscape maps can be valuable tools for stakeholders in environmental noise management and urban planning [18, 29, 31]. Additionally, various commercial technologies, such as augmented/virtual reality and navigation systems, can utilize soundscape mapping to provide an immersive experience.

Traditionally, soundscape mapping has been formulated as learning a predictive model that maps a fixed set of acoustic indicators

Permission to make digital or hard copies of all or part of this work for personal or classroom use is granted without fee provided that copies are not made or distributed for profit or commercial advantage and that copies bear this notice and the full citation on the first page. Copyrights for components of this work owned by others than the author(s) must be honored. Abstracting with credit is permitted. To copy otherwise, or republish, to post on servers or to redistribute to lists, requires prior specific permission and/or a fee. Request permissions from permissions@acm.org.
*ACM MM, 2024, Melbourne, Australia*
© 2024 Copyright held by the owner/author(s). Publication rights licensed to ACM.
ACM ISBN 978-x-xxxx-xxxx-x/YY/MM
https://doi.org/10.1145/nnnnnnn.nnnnnnn

(such as sound pressure, loudness, etc.) to a fixed set of descriptors (such as pleasant, eventful, etc.) [15, 17, 27]. However, this abstraction prevents us from fully understanding the underlying acoustic scene at a location. Moreover, soundscape maps created in such a manner rely on crowd-sourced data [5, 34], which is often available only for densely populated and highly visited locations. Therefore, traditional soundscape mapping techniques can only generate sparse soundscape maps that lack generalizability beyond regions with sufficient data. Consequently, these techniques are not suitable for creating dense global soundscape maps.

To address the limitations of traditional soundscape mapping, we adopt a formulation where, given a specific location, the task is to train a machine learning model that directly predicts the sound distribution likely to be encountered at that location. We represent each location with a satellite image centered around it. This approach enables the generalization of soundscape mapping beyond locations explicitly included in the training data.

We approach the soundscape mapping problem from the perspective of multimodal representation learning to design a shared embedding space between audio and satellite imagery at the recorded location of the audio. This learning strategy aims to bring positive audio-satellite image pairs closer while pushing negative pairs farther apart in the embedding space. Ultimately, the multimodal embedding space can be employed to generate soundscape maps by computing similarity scores between the query and the satellite image set covering the geographic region of interest.

However, the problem of soundscape mapping is inherently uncertain. In most cases, multiple types of sounds can come from a given geographic location. Similarly, a specific type of sound can also come from multiple geographic locations. As such, paired location and audio data are assured to contain sample pairs that are labeled as negatives but are semantically similar to positives. We call such sample pairs as pseudo-positives. Any method that learns completely deterministic representations of sound and satellite imagery of the location of the sound ignores the uncertainty involved in soundscape mapping. To address this, we argue that learning a probabilistic cross-modal embedding space is more suitable for this task. Accordingly, we learn a probabilistic multi-modal embedding space [11] between audio, satellite imagery, and textual description of audio. Moreover, to account for potential false negative matches during mapping, we identify pseudo-positive matches during training [11].

The satellite image representing the capture location of the audio can be obtained at different spatial resolutions, where the ground area coverage of the satellite image increases as the zoom level increases. In our work to create large-scale soundscape maps, we are interested in learning an embedding space that models differences in the spatial resolution of the satellite imagery. Therefore, we modify the zero-shot soundscape mapping formulation as multi-scale zero-shot soundscape mapping so that ground-level sounds

may be mapped with satellite imagery at different zoom levels. We achieve this by learning a shared satellite image encoder across different zoom levels that utilize a recently proposed Ground-Sample Distance Positional Embedding (GSDPE) [36].

Our modalities of interest, satellite imagery, audio, and text, often have associated metadata that convey meaningful information (such as latitude and longitude or the source of an audio sample). We propose to fuse such metadata: location, time, and source from which the audio was collected, into our framework. We demonstrate that such information increases the discriminative power of our embedding space and allows the creation of soundscape maps conditioned on dynamic metadata settings during inference.

The most closely related prior work [25] in soundscape mapping was trained on limited data ($\sim$ 35k samples) from the *SoundingEarth* dataset [20]. To advance research in this area, we curated a new large-scale dataset, *GeoSound*, by collecting geotagged audios from four different sources, increasing the dataset size six-fold to over 300k samples. We use *GeoSound* to train our framework that advances the state-of-the-art in zero-shot soundscape mapping by learning a probabilistic, scale-aware, and metadata-aware joint multimodal embedding space. Moreover, we demonstrate the capability of the proposed framework in the creation of temporally dynamic soundscape maps.

The main contributions of our work are as follows:

- We introduce a new large-scale dataset containing over 300k geotagged audios paired with high-resolution (0.6m) and low-resolution (10m) satellite imagery.
- We learn a metadata-aware, probabilistic embedding space between satellite imagery, audio, and textual audio description for zero-shot multi-scale soundscape mapping.
- We demonstrate the utility of our framework (PSM: **P**robabilistic **S**oundscape **M**apping) in creating large-scale soundscape maps created by querying our learned embedding space with audio or text.

## 2 RELATED WORKS

### 2.1 Audio Visual Learning

An intricate relationship exists between the audio and visual attributes of a scene. Utilizing this relationship, there have been numerous works in the field of audio-visual learning. [9, 20, 21, 23, 25, 33, 37, 45, 46]. Owens *et al.* [33] have demonstrated that encouraging the models to predict sound characteristics of a scene allows them to learn richer representations useful for visual recognition tasks. Hu *et al.* [21] proposed to learn from audio and images to solve the task of aerial scene recognition. Relatively closer to the formulation of our work, Salem *et al.* [37] proposed to learn a shared feature space between satellite imagery, ground-level concepts, and audio, which allowed them to predict sound cluster distribution across large geographic regions. Recently, Khanal *et al.* [25] proposed the learning of a tri-modal embedding space to map satellite imagery with the most likely audio at a location.

### 2.2 Deterministic Contrastive Learning

The contrastive learning paradigm [28, 35, 39, 42] has significantly advanced state-of-the-art multimodal learning capabilities through rich cross-modal supervision. In the pursuit of advancing contrastive learning approaches for audio and text, Elizalde et al. [14] and Wu et al. [44] have developed a Contrastive Language-Audio Pretraining (CLAP) framework, showcasing strong zero-shot capabilities. Wav2CLIP [43] distills information learned from CLIP to create a joint image-audio embedding space. AudioCLIP [19] extends contrastive learning to audio, image, and text, exhibiting impressive performance across various downstream tasks. Recently, Heidler et al. proposed learning a shared representation space between audio and corresponding satellite imagery for use in various downstream tasks in remote sensing. Similarly, Khanal et al. [25] utilized the *SoundingEarth* dataset [20] to train a multimodal embedding space using a deterministic contrastive loss [32] for zero-shot soundscape learning.

### 2.3 Probabilistic Contrastive Learning

In our formulation of soundscape mapping, the satellite image provided as location context captures a geographic area containing many sound sources. As such, deterministic contrastive learning approaches cannot capture the inherent ambiguity in the mapping from satellite image to sound, as any sample can only be represented by a single point in the embedding space. This limitation can be addressed by representing embeddings probabilistically [7, 8, 11, 12, 22, 24, 30, 38, 41]. In other words, each sample in probabilistic embedding space is represented by a probability distribution whose parameters are learned, usually by a neural network. A work by Chun *et al.*, Probabilistic Cross-Modal Embeddings (PCME) [12], represents samples as Gaussian distributions in the embedding space and trains their framework using a contrastive loss between the sample distributions computed by Monte-Carlo sampling. Recently, Chun [11] proposed PCME++, which further improved PCME by introducing a closed-form distance formulation that removes the need for Monte-Carlo sampling to approximate distribution differences. In our work, we adopt the PCME++ embedding formulation to learn a probabilistic embedding space between audio, a textual description of audio, and multiscale satellite imagery at the location of audio.

## 3 METHOD

This section describes the novel dataset we curated and our proposed framework, PSM.

### 3.1 Dataset Creation

Prior work in zero-shot soundscape mapping [25] has utilized the *SoundingEarth* dataset [20], which contains approximately 50k geotagged audios paired with corresponding satellite imagery. To facilitate research on training large-scale models with a rich representation space for soundscape mapping, we have expanded the size of the dataset 6-fold by creating a dataset containing 309 019 geo-tagged audios from four different sources: *iNaturalist* [3], *yfcc-video* [40], *aporee* [4], and *freesound* [1], each contributing 114 603, 96 452, 49 284, and 48 680 samples respectively. We pair these geo-tagged audios with their corresponding *Sentinel-2-cloudless* imagery with 10m GSD and 0.6m GSD *Bing* imagery.

In the prior work, GeoCLAP [25], samples were randomly split between train/validation/test sets for training and evaluating their

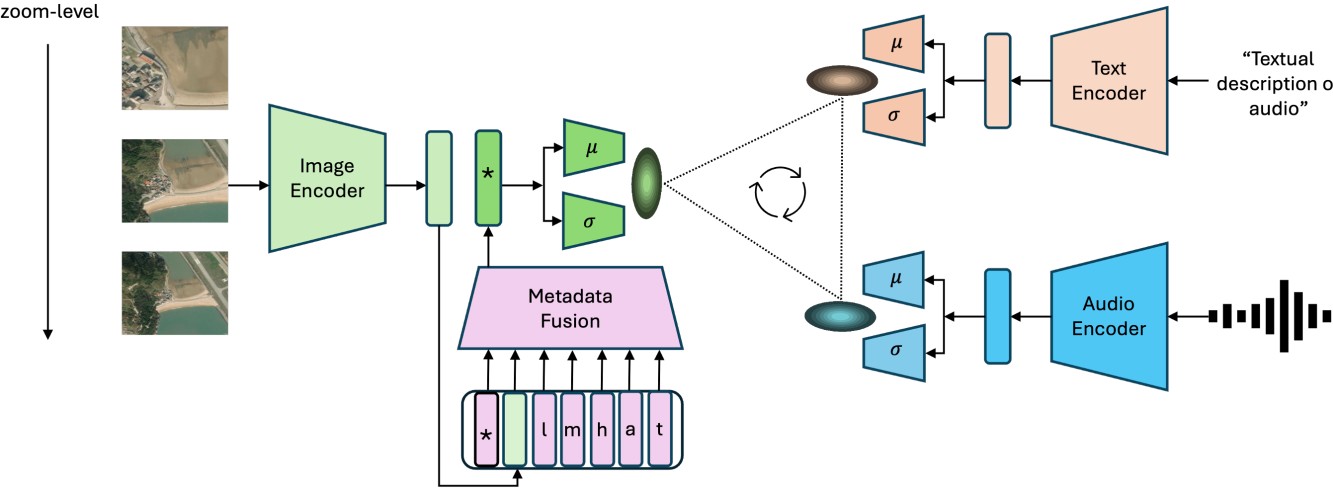

**Figure 1: Our proposed framework, Probabilistic Soundscape Mapping (PSM), combines image, audio, and text encoders to learn a probabilistic joint representation space. Metadata, including geolocation (l), month (m), hour (h), audio-source (a), and caption-source (t), is encoded separately and fused with image embeddings using a transformer-based metadata fusion module. For each encoder, $\mu$ and $\sigma$ heads yield probabilistic embeddings, which are used to compute probabilistic contrastive loss.**

models. We observed that such a data split strategy leads to the issue of data leakage where evaluation data samples come from the same set of locations present in the training set, preventing the evaluation of the generalizability of a model to truly unseen locations. To address this, we divide the world into $1° \times 1°$ non-overlapping cells where each cell containing some samples is randomly assigned to either train/validation/test set. Our dataset contains $294\,019/5000/10\,000$ samples in the train/validation/test sets. We also employ our split strategy on the *SoundingEarth* dataset with a cell size of $10km \times 10km$. This strategy resulted in $41\,469/3242/5801$ samples in train/validation/test sets. Details of our dataset and split strategy are in the supplemental material.

### 3.2 Approach

This section describes our framework (PSM) for learning a metadata-aware, probabilistic, and tri-modal embedding space for multi-scale zero-shot soundscape mapping.

Figure 1 presents an overview of the PSM framework, which comprises an image encoder, metadata fusion module, text encoder, and audio encoder. The scale-aware image encoder converts multiscale satellite imagery into a $d$-dimensional representation. The transformer-based metadata fusion module integrates metadata (including location, month, time, audio source, and text source) with the image representation, generating a metadata-aware probabilistic image representation. Other modality-specific encoders produce probabilistic embeddings for text and audio. PSM aims to map tuples of satellite imagery, audio, and text into a shared probabilistic representation space.

Given a geotagged audio $X_k^a$, textual description of the audio $X_k^t$, and a satellite image at a given location viewed at a zoom level $l$ (an integer between 1 and some maximum zoom level $L$) $X_{k,l}^i$, $(X_k^a, X_k^t, X_{k,l}^i)$ is the $k$-th audio-text-image triplet. PSM is trained over the aggregation of all available triplets.

We use modality-specific transformer-based encoders followed by their respective linear projection layers to obtain representations $(h_k^a, h_k^t, h_{k,l}^i)$ with same dimension $d$.

$$h_k^a = g_{audio}(f_{audio}(X_k^a)) \qquad (1)$$

$$h_k^t = g_{text}(f_{text}(X_k^t)) \qquad (2)$$

$$h_k^i = g_{image}(f_{image}(X_k^i, l_k)) \qquad (3)$$

where $(f_{audio}, g_{audio})$, $(f_{text}, g_{text})$, $(f_{image}, g_{image})$ are (encoder, projection-module) pairs producing $d$ dimensional embeddings: $h_k^a$, $h_k^t$, and $h_k^i$, for audio, text, and satellite image with zoom-level $l_k$ respectively.

We use GSDPE [36] to encode the position and scale of each patch of satellite imagery at zoom-level ($l$) to learn scale-aware representations of multiscale satellite imagery,

$$v_{l,x}(pos, 2i) = sin(\frac{g*l}{G}) \frac{pos}{10000^{\frac{2i}{d}}} \qquad (4)$$

$$v_{l,y}(pos, 2i+1) = cos(\frac{g*l}{G}) \frac{pos}{10000^{\frac{2i}{d}}} \qquad (5)$$

where $pos$ is the position of the image patch along the given axis ($x$ or $y$), $i$ is the feature dimension index, $l$ is the zoom-level of the image, $g$ is the GSD of the image, and $G$ is the reference GSD.

As discussed before, we are interested in learning metadata-aware representation space. Therefore, we fuse four different components of metadata (geolocation, month, hour, audio-source, caption-source) with the satellite image embedding ($h_k^i$) and obtain a metadata-conditioned image embedding ($h_k^{i'}$).

$$h_k^{i'} = g_{meta}(h_k^i, metadata) \qquad (6)$$

where $g_{meta}$ is the metadata fusion module of our framework, $h_k^{i'}$ is the embedding corresponding to the learnable special token (*) fed into $g_{meta}$.

To learn a probabilistic embedding space, we define the embedding of a given modality ($r$) as a normally distributed random variable, $Z_r \sim N(\mu_r, \sigma_r)$. We employ a closed-form probabilistic contrastive loss [11] between all three pairs of embeddings. For any two modalities $p$ and $q$, the closed-form sampled distance (CSD) as defined in PCME++ [11] is:

$$d(Z_p, Z_q) = \|\mu_p - \mu_q\|_2^2 + \|\sigma_p^2 + \sigma_q^2\|_1 \tag{7}$$

In our implementation, we pass our modality-specific representations, $h_k^a$, $h_k^t$, and $h_k^{i'}$, through heads for $\mu$ and $\log(\sigma^2)$ of the Gaussian distribution representing our samples.

Based on the distance function defined in Equation 7, we can then define the probabilistic matching objective function as follows:

$$\mathcal{L}_m = -w_{pq} \log(\text{sigmoid}(-a.d(Z_p, Z_q) + b)) - \\ (1 - w_{pq}) \log(\text{sigmoid}(a.d(Z_p, Z_q) - b)) \tag{8}$$

where $w_{pq} \in \{0, 1\}$ is the matching indicator between $p$ and $q$. $a$ and $b$ are learnable scalar parameters. $\mathcal{L}_m$ ($\mathcal{L}_{match}$) is computed for all sample pairs in the mini-batch.

Soundscape mapping is inherently a one-to-many matching problem. Given a satellite image at a location, there may be multiple sounds that are likely to be heard there. Therefore, if we were to simply assign $w_{pq}$ as 0 or 1 for our dataset's negative and positive matches, we would lose the opportunity to learn from the potentially numerous false negatives. Therefore, we adopt a similar strategy of learning from pseudo-positives, as formulated by Chun [11]. In this approach, for a positive match ($p$,$q$), we consider $q'$ as a pseudo-positive match with $q$ if $d(Z_p, Z_{q'}) \leq d(Z_p, Z_q)$. Finally, the objective function for a pair of modalities ($p$, $q$) becomes as follows:

$$\mathcal{L}_{p,q} = \mathcal{L}_m + \alpha \mathcal{L}_{pseudo-m} + \beta \mathcal{L}_{VIB} \tag{9}$$

where $\alpha$ and $\beta$ control for the contribution of pseudo-match loss and Variational Information Bottleneck (VIB) loss [6], respectively. We use $\mathcal{L}_{VIB}$ as a regularizer to reduce overfitting, preventing the collapse of $\sigma$ to 0.

To learn a tri-modal embedding space for zero-shot soundscape mapping, using Equation 9, we separately compute loss for all three pairs of modalities: audio-text ($a, t$), audio-image ($a, i$), and image-text($i, t$). Finally, the overall objective function to train PSM is as follows:

$$\mathcal{L} = \mathcal{L}_{a,t} + \mathcal{L}_{a,i} + \mathcal{L}_{i,t} \tag{10}$$

## 4 EXPERIMENTAL DETAILS

**Audio/Text Processing:** We use pre-trained models for the audio and text modalities and their respective input processing pipelines hosted on HuggingFace. Specifically, for audio, we extract the audio spectrogram using the ClapProcessor wrapper for the pretrained CLAP [44] model clap-htsat-fused with default parameters: feature_size=64, sampling_rate=48000, hop_length=480, fft_window_size=1024. CLAP uses a feature fusion strategy [44] to pre-process variable length sounds by extracting a spectrogram of randomly selected 3 $d$-second audio slices and the spectrogram of the whole audio down-sampled to 10s. We choose $d$ =10s in our experiments. Apart from the text present in the metadata, we also obtain a textual description of audio from a recent SOTA audio captioning model, Qwen-sound [10], and use the captioning model's

output only if it passes CLAP-score [44] based quality check. For the textual descriptions of audio in our data, we adopt the similar text processing as performed by CLAP [44] and tokenize our text using RobertaTokenizer with max_length=128.

**Satellite image processing:** Our framework is trained with satellite images at different zoom levels $l \in \{1, 3, 5\}$. To obtain this data, we first downloaded a large tile of images with size $(Lh) \times (Lw)$. We obtained high-resolution 0.6m GSD imagery with a tile size of $1500 \times 1500$ from *Bing* and low-resolution 10m GSD imagery with a tile size of $1280 \times 1280$ from *Sentinel-2-cloudless*. To get an image at zoom-level $l$, we center crop from the original tile with a crop size of $(lh) \times (lw)$ and then resize it to an $h \times w$ image, where $(h, w)$ is $(256, 256)$ for *Sentinel-2* imagery and $(300, 300)$ for *Bing* imagery. This way, we can simulate the effect of change in coverage area as the zoom-level changes while effectively keeping constant input image size for training. During training, we randomly sample $l$ from a set $\{1, 3, 5\}$ for each image instance. Then, for the zoom-transformed image, we perform *RandomResizedCrop* with parameters: {input_size=224, scale=(0.2, 1.0)} followed by a *RandomHorizontalFlip* while only extracting a $224 \times 224$ center crop of the image at the desired zoom-level $l$ for evaluation.

**Metadata Fusion:** To fuse metadata into our framework, we first separately project the metadata components into 512-dimensional space using linear layers and concatenate them with the satellite image embedding from the image encoder and a learnable special token. Finally, the set of tokens is fed into a lightweight transformer-based module containing only 3 layers. The output of this module is further passed through heads for $\mu$ and $\log(\sigma^2)$ of the Gaussian distribution representing metadata-conditioned image embeddings. To avoid overfitting to the metadata, we independently drop each metadata component at the rate of 0.5 during training.

**Training:** We initialize encoders from released weights of pre-trained models, SatMAE [13] for satellite imagery and CLAP [44] for audio and text. We chose $d$, the dimensionality of our embeddings, to be 512. For regularization, we set the weight decay to 0.2. Our training batch_size was 128. We use Adam as our optimizer, with the initial learning rate set to $5e - 5$. To schedule the learning rate, we use cosine annealing with warm-up iterations of $5k$ for experiments with *GeoSound* and $2k$ for experiments with *SoundingEarth*.

**Baseline:** We use GeoCLAP [25], a SOTA zero-shot soundscape mapping model, as a baseline for evaluation. GeoCLAP is contrastively trained using the *infoNCE* [32] loss between three modality pairs: image-audio, audio-text, and image-text.

**Metrics:** We evaluate on two datasets: *GeoSound*, and *SoundingEarth*. We use Recall@10% and the Median Rank of the ground truth as our evaluation metrics. Recall@10% is defined as the proportion of queries that include the ground-truth match in the top 10% of the returned ranked retrieval list. We denote image-to-audio as I2A and audio-to-image as A2I throughout the paper. Median Rank is defined as the median overall positions in which the ground-truth match appears in the ranked retrieval list. To assess the effectiveness of text embeddings in cross-modal retrieval between satellite images and audio, we also evaluate an experimental setting where, during inference, we add the corresponding text embedding to the query embedding during retrieval from the respective gallery.

| method | loss | text | metadata | zoom level | I2A R@10% | I2A median rank | A2I R@10% | A2I median rank |
|---|---|---|---|---|---|---|---|---|
| GeoCLAP | infoNCE | ✗ | ✗ | 1 | 0.399 | 1500 | 0.403 | 1464 |
| GeoCLAP | infoNCE | ✓ | ✗ | 1 | 0.577 | 712 | 0.468 | 1141 |
| ours | infoNCE | ✓ | ✓ | 1 | 0.709 | 462 | 0.871 | 241 |
| ours | PCME++ | ✗ | ✗ | 1 | 0.423 | 1401 | 0.428 | 1344 |
| ours | PCME++ | ✗ | ✓ | 1 | 0.828 | 261 | 0.829 | 248 |
| ours | PCME++ | ✓ | ✓ | 1 | **0.901** | **113** | **0.943** | **100** |
| GeoCLAP | infoNCE | ✗ | ✗ | 3 | 0.408 | 1441 | 0.420 | 1389 |
| GeoCLAP | infoNCE | ✓ | ✗ | 3 | 0.577 | 707 | 0.483 | 1056 |
| ours | infoNCE | ✓ | ✓ | 3 | 0.708 | 462 | 0.875 | 235 |
| ours | PCME++ | ✗ | ✗ | 3 | 0.440 | 1302 | 0.443 | 1266 |
| ours | PCME++ | ✗ | ✓ | 3 | 0.827 | 266 | 0.832 | 250 |
| ours | PCME++ | ✓ | ✓ | 3 | **0.900** | **114** | **0.945** | **102** |
| GeoCLAP | infoNCE | ✗ | ✗ | 5 | 0.409 | 1428 | 0.421 | 1373 |
| GeoCLAP | infoNCE | ✓ | ✗ | 5 | 0.581 | 698 | 0.489 | 1036 |
| ours | infoNCE | ✓ | ✓ | 5 | 0.709 | 461 | 0.875 | 238 |
| ours | PCME++ | ✗ | ✗ | 5 | 0.440 | 1302 | 0.448 | 1279 |
| ours | PCME++ | ✗ | ✓ | 5 | 0.821 | 281 | 0.826 | 261 |
| ours | PCME++ | ✓ | ✓ | 5 | **0.896** | **115** | **0.941** | **107** |

**Table 1: Experimental results for models trained on the GeoSound dataset with satellite imagery from *Bing*.**

| method | loss | text | metadata | zoom level | I2A R@10% | I2A median rank | A2I R@10% | A2I median rank |
|---|---|---|---|---|---|---|---|---|
| GeoCLAP | infoNCE | ✗ | ✗ | 1 | 0.459 | 1179 | 0.465 | 1141 |
| GeoCLAP | infoNCE | ✓ | ✗ | 1 | 0.546 | 827 | 0.553 | 804 |
| ours | infoNCE | ✓ | ✓ | 1 | 0.722 | 497 | 0.86 | 247 |
| ours | PCME++ | ✗ | ✗ | 1 | 0.474 | 1101 | 0.485 | 1061 |
| ours | PCME++ | ✗ | ✓ | 1 | 0.802 | 294 | 0.804 | 283 |
| ours | PCME++ | ✓ | ✓ | 1 | **0.872** | **142** | **0.940** | **104** |
| GeoCLAP | infoNCE | ✗ | ✗ | 3 | 0.454 | 1200 | 0.456 | 1197 |
| GeoCLAP | infoNCE | ✓ | ✗ | 3 | 0.542 | 840 | 0.555 | 790 |
| ours | infoNCE | ✓ | ✓ | 3 | 0.722 | 491 | 0.856 | 248 |
| ours | PCME++ | ✗ | ✗ | 3 | 0.479 | 1086 | 0.487 | 1042 |
| ours | PCME++ | ✗ | ✓ | 3 | 0.795 | 306 | 0.800 | 290 |
| ours | PCME++ | ✓ | ✓ | 3 | **0.870** | **150** | **0.940** | **104** |
| GeoCLAP | infoNCE | ✗ | ✗ | 5 | 0.458 | 1194 | 0.457 | 1184 |
| GeoCLAP | infoNCE | ✓ | ✗ | 5 | 0.542 | 835 | 0.554 | 791 |
| ours | infoNCE | ✓ | ✓ | 5 | 0.719 | 497 | 0.852 | 252 |
| ours | PCME++ | ✗ | ✗ | 5 | 0.459 | 1172 | 0.465 | 1138 |
| ours | PCME++ | ✗ | ✓ | 5 | 0.794 | 316 | 0.794 | 299 |
| ours | PCME++ | ✓ | ✓ | 5 | **0.868** | **156** | **0.935** | **109** |

**Table 2: Experimental results for models trained on the GeoSound dataset with satellite imagery from *Sentinel-2*.**

## 5 RESULTS

In this section, we discuss the experimental results of our framework, PSM, over separate training with *Sentinel-2* and *Bing* imagery of *GeoSound* dataset as well as on *SoundingEarth* dataset. We evaluate our models for cross-modal retrieval performance between satellite imagery and audio. We also display soundscape maps created by querying our framework with audio or text.

 

| method | loss | text | metadata | I2A R@10% | I2A median rank | A2I R@10% | A2I median rank |
|--------|------|------|----------|-----------|-----------------|-----------|-----------------|
| GeoCLAP | infoNCE | ✗ | ✗ | 0.454 | 667 | 0.449 | 694 |
| GeoCLAP | infoNCE | ✓ | ✗ | 0.523 | 533 | 0.470 | 641 |
| ours | infoNCE | ✓ | ✓ | 0.519 | 548 | 0.491 | 596 |
| ours | PCME++ | ✗ | ✗ | 0.514 | 547 | 0.518 | 543 |
| ours | PCME++ | ✗ | ✓ | 0.563 | 454 | 0.569 | 447 |
| ours | PCME++ | ✓ | ✓ | **0.690** | **264** | **0.608** | **371** |

**Table 3: Experimental results for models trained on the SoundingEarth dataset with satellite imagery from *GoogleEarth*.**

| imagery | latlong | month | time | audio source | text source | I2A R@10% | I2A median rank | A2I R@10% | A2I median rank |
|---------|---------|-------|------|--------------|-------------|-----------|-----------------|-----------|-----------------|
| Sentinel-2 | ✓ | ✗ | ✗ | ✗ | ✗ | 0.512 | 946 | 0.516 | 923 |
| Sentinel-2 | ✗ | ✓ | ✗ | ✗ | ✗ | 0.501 | 988 | 0.511 | 941 |
| Sentinel-2 | ✗ | ✗ | ✓ | ✗ | ✗ | 0.548 | 825 | 0.574 | 717 |
| Sentinel-2 | ✗ | ✗ | ✗ | ✓ | ✗ | **0.749** | **407** | **0.757** | **389** |
| Sentinel-2 | ✗ | ✗ | ✗ | ✗ | ✓ | 0.483 | 1080 | 0.492 | 1022 |
| Bing | ✓ | ✗ | ✗ | ✗ | ✗ | 0.539 | 822 | 0.557 | 764 |
| Bing | ✗ | ✓ | ✗ | ✗ | ✗ | 0.464 | 1153 | 0.485 | 1068 |
| Bing | ✗ | ✗ | ✓ | ✗ | ✗ | 0.516 | 937 | 0.547 | 823 |
| Bing | ✗ | ✗ | ✗ | ✓ | ✗ | **0.722** | **469** | **0.733** | **447** |
| Bing | ✗ | ✗ | ✗ | ✗ | ✓ | 0.448 | 1250 | 0.466 | 1140 |

**Table 4: Metadata Ablation to evaluate the impact of individual metadata components on the best model's performance.**

## 5.1 Cross-Modal Retrieval with Bing

Table 1 presents our retrieval evaluation of PSM trained on the *GeoSound* dataset using *Bing* satellite imagery. Our approach outperforms the state-of-the-art baseline [25] for cross-modal retrieval between satellite imagery and audio, and vice versa. SatMAE [13] with GSDPE is utilized to encode the zoom level of the satellite imagery for both the baseline and our models. This enables our satellite image encoder to remain invariant to zoom-level changes, achieving consistent performance across all zoom levels. We observe that learning a probabilistic embedding space using PCME++ loss alone enhances the baseline performance from 0.399 to 0.423, 0.408 to 0.440, and 0.409 to 0.440 for zoom levels 1, 3, and 5, respectively. In addition to the objective function, we also experimented with the inclusion of metadata during training and inference. As anticipated, the model's performance, when trained and evaluated with both text and metadata, is notably improved, enhancing image-to-audio retrieval @ 10% from the baseline score of 0.577 to 0.901, 0.577 to 0.900, and 0.581 to 0.896 for zoom levels 1, 3, and 5, respectively. A similar trend is observed for audio-to-image retrieval.

## 5.2 Cross-Modal Retrieval with Sentinel-2

Table 2 presents the evaluation results of PSM trained on the *GeoSound* dataset using *Sentinel-2* satellite imagery. Similar to experiments with *Bing* imagery, we observe consistent performance across various zoom levels, indicating the robustness of our framework in extracting valuable information irrespective of the coverage area of input satellite imagery. By employing PCME++ loss in training our framework, we note an enhancement in the baseline performance from 0.459 to 0.474 for zoom level 1. Overall, PSM

trained with *Sentinel-2* imagery and metadata, and evaluated using both metadata and text during inference, significantly improved the baseline score from 0.546 to 0.872, 0.542 to 0.870, and 0.542 to 0.868 for zoom levels 1, 3, and 5, respectively. A similar trend is observed for audio-to-image retrieval. The high performance of PSM on *Sentinel-2* imagery at zoom level 5 enables the efficient creation of large-scale soundscape maps utilizing freely available *Sentinel-2* imagery.

## 5.3 Cross-Modal Retrieval on SoundingEarth

Table 3 presents the evaluation results of PSM trained on the *SoundingEarth* dataset [20] with its original 0.2m GSD *GoogleEarth* imagery. For the *SoundingEarth* dataset, our models are exclusively trained and evaluated on zoom level 1. Similar to the performance observed on the *GeoSound* dataset, we witness gain in performance with our approach of learning a metadata-aware probabilistic embedding space. Specifically, by training with the PCME++ objective instead of the *infoNCE* loss, we note an improvement in the score from 0.454 to 0.514. This performance further elevates to 0.563 when metadata is incorporated and reaches 0.690 when both metadata and text are utilized during inference. We observe similar trends for audio-to-image retrieval as well.

## 5.4 Effect of Metadata

Our experimental results reveal a significant enhancement in the model's performance when metadata is integrated into both training and inference. For comparison, as illustrated in Table 1, PSM trained with *Bing* imagery without any metadata achieved an I2A R@10% of 0.423, whereas with all metadata included, it reached

Text query: Sound of insects

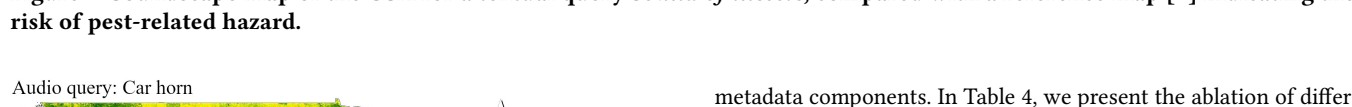

**Figure 2: Soundscape Map of the USA for a textual query *Sound of insects,* compared with a reference map [2] indicating the risk of pest-related hazard.**

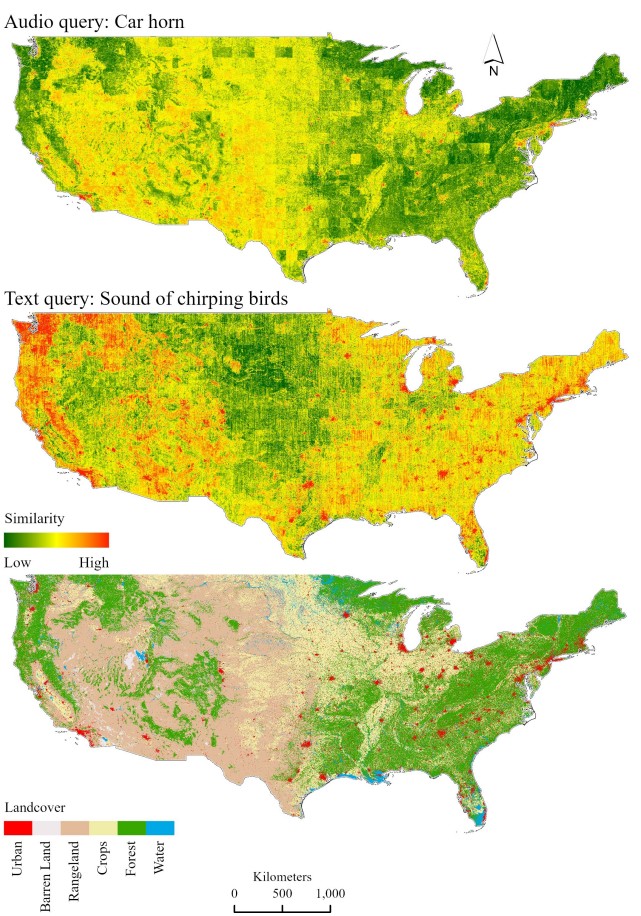

**Figure 3: Two soundscape maps of the continental United States, generated using different query types, with a land cover map [16] for reference.**

0.828. A similar trend is seen for experiments with *Sentinel-2* imagery. PSM is designed such that individual metadata components are independently masked out with a rate of 0.5. Therefore, during inference, we can evaluate PSM by dropping any combination of

metadata components. In Table 4, we present the ablation of different metadata components to evaluate the impact of individual metadata components in PSM's learning framework. We conduct this ablation on our best-performing models trained on the *GeoSound* dataset with both satellite imagery types: *Sentinel-2* and *Bing*. All ablation experiments are conducted on imagery with zoom level 1. The results reported in Table 4 do not involve the use of text during cross-modal retrieval.

As observed in the ablation results, for the best-performing model trained with *Sentinel-2* imagery, the performance due to the addition of text source slightly increases from 0.474 to 0.483. However, this performance increases to 0.501, 0.512, 0.548, and 0.749 when the model is evaluated with the independent addition of other metadata components: month, latlong, time, and audio source, respectively. Similarly, for a model trained with *Bing* imagery, the performance due to the addition of text source slightly increases from 0.423 to 0.448. However, this performance increases to 0.464, 0.516, 0.539, and 0.722 when the model is evaluated with the independent addition of other metadata components: month, time, latlong, and audio source, respectively. These results highlight two major findings. First, all of the metadata components contribute to the overall improvement of PSM's performance. Second, among all of the metadata components, audio-source had the most significant impact. This suggests that the inherent biases present in different audio data hosting platforms were explicitly encoded into the learning framework. This facilitates not only the improvement of cross-modal retrieval performance but also enables the creation of soundscape maps conditioned on the type of audio expected to be found in a specific audio data hosting platform.

## 5.5 Generating Country-level Soundscape Maps

We demonstrate PSM's capability to generate large-scale soundscape maps using audio and text queries. We acquired 0.6 m GSD 1500 × 1500 image tiles encompassing the entire USA from *Bing*. Employing our top-performing model's image encoder, we precomputed embeddings for each image at zoom-level 1. During inference, these pre-computed embeddings are combined with desired metadata embeddings using the model's metadata fusion module to get metadata-conditioned probabilistic embeddings for the entire region. We leverage modality-specific encoders of the model to get probabilistic embeddings for audio or text queries. Finally, to

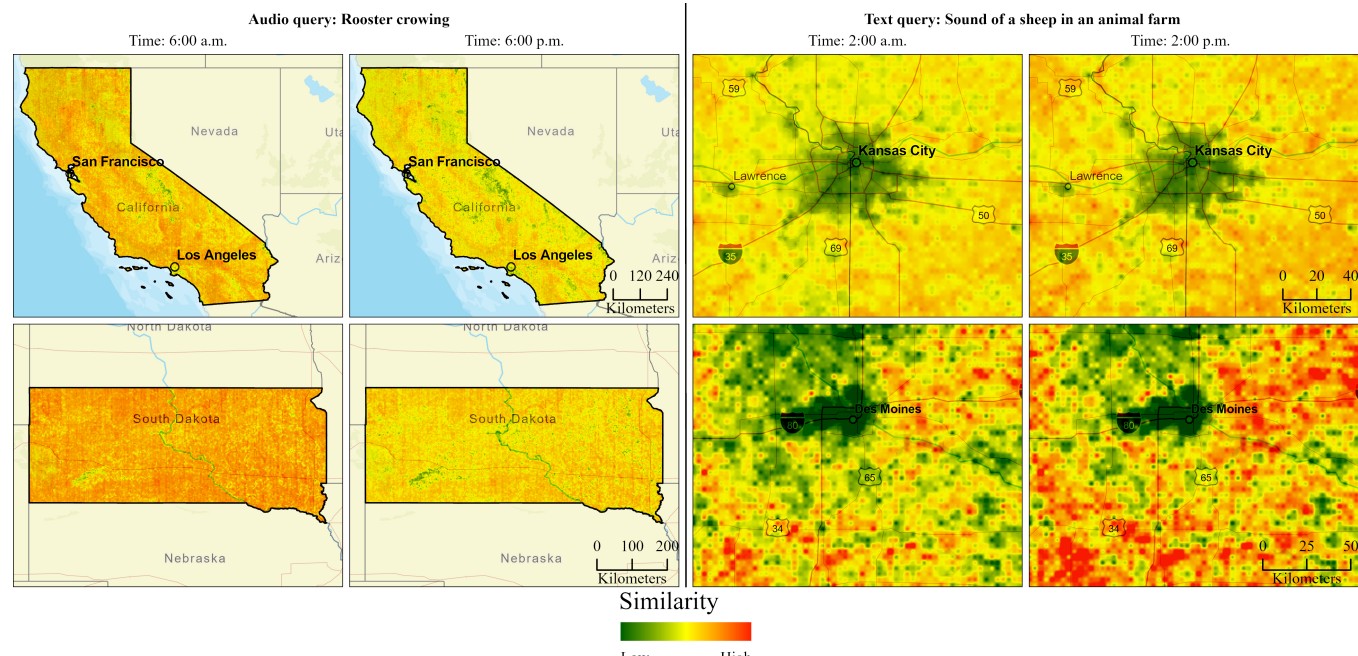

**Figure 4: Temporally dynamic soundscape maps created by querying our model for different geographic areas.**

compute the similarity score of all image embeddings for the region with the probabilistic embeddings for the query, we utilize Equation 7 as detailed in our paper. Subsequently, these similarity scores are used to produce large-scale soundscape maps, as illustrated in Figures 2, and 3.

## 6 DISCUSSION

Figure 2 depicts a soundscape map generated for the textual query *"Sound of insects"*, accompanied by the following metadata: {audio source: iNaturalist, month: May, time: 8 pm}. Notably, this soundscape map exhibits a strong correlation with an available reference map [2], which shows potential pest hazards across the continental United States. Figure 3 showcases two soundscape maps: one for an audio query of *car horn* with the metadata {audio source: yfcc, month: May, time: 10 am}, and another for a textual query *"Sound of chirping birds."* with metadata: {audio source: iNaturalist, month: May, time: 10 am}. Both maps can be compared with a land cover map [16]. As expected, for the car horn query, higher activation is observed in most major US cities, while for chirping birds, increased activation is observed around both urban areas and forests.

We also note that the soundscape of any geographic region evolves predictably over the course of a day. Therefore, the hour of the day is one of the important metadata components fused into our framework. In addition to contributing to increased performance, temporal understanding fused into our embedding space allows us to create temporally dynamic soundscape maps across any geographic region, as demonstrated in Figure 4. The similarity scores used for these soundscape maps were normalized consistently for a region across time. We display state-level temporally dynamic

soundscape maps for an audio query: *Rooster crowing* with metadata: {audio source: aporee, month: May, time: 6 am} vs. {audio source: aporee, month: May, time: 6 pm}. We observe that for both states, higher activation for the rooster crowing audio query is seen on the soundscape map at 6 am. We also showcase city-level temporally dynamic soundscape maps for a text query *"Sound of a sheep in an animal farm."* We can observe that for areas around both cities, Kansas City and Des Moines, there is very low activation. In addition, higher activation is observed at 2 pm than at 2 am, which is expected. These demonstrations highlight the ability of our model to create semantically meaningful and temporally consistent soundscape maps.

## 7 CONCLUSION

Our work introduces a framework for learning probabilistic tri-modal embeddings for the task of multi-scale zero-shot soundscape mapping. To advance research in this direction, we have developed a new large-scale dataset that pairs geotagged audio with high and low-resolution satellite imagery. By utilizing a probabilistic tri-modal embedding space, our method surpasses the state-of-the-art while also providing uncertainty estimates for each sample. Furthermore, we have designed our framework to be metadata-aware, resulting in a significant improvement in cross-modal retrieval performance. Additionally, it enables the creation of dynamic soundscape maps conditioned on different types of metadata. The combination of enhanced mapping performance, uncertainty estimation, and a comprehensive understanding of spatial and temporal dynamics positions our framework as an effective solution for zero-shot multi-scale soundscape mapping.

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
