# OpenReview forum: "PSM: Learning Probabilistic Embeddings for Multi-scale Zero-shot Soundscape Mapping"
_acmmm.org/ACMMM/2024/Conference — MM2024 Poster_

### Official Review · Reviewer_Uedh · 2024-05-16

**Rating:** 5
**Confidence:** 2

**Summary:**

This paper presents a framework for mapping global soundscapes using probabilistic embeddings from multi-scale satellite imagery, audio, and text data. The study introduces a novel method that integrates metadata like geolocation and time to enhance the model's ability to predict and understand sound distributions across diverse geographic locations. A key advancement is the development of the GeoSound dataset, which contains over 300,000 geotagged audio samples. The proposed approach outperforms existing methods by effectively capturing the inherent uncertainty in soundscape data, providing a robust tool for environmental monitoring and urban planning.

**Strengths:**

Novelty: The framework introduces a probabilistic approach to soundscape mapping, addressing the inherent variability and uncertainty of environmental sounds; it also proposes a new dataset.
Theoretical Approach: Using probabilistic embeddings to capture the complexity of soundscapes may be theoretically reasonable.
Evaluation: The performance improvement is demonstrated through comparisons with existing methods on two datasets, highlighting the effectiveness of the proposed method.
Clarity: The structure of the paper is relatively clear.

**Limitations:**

Data and Metadata Dependency: The effectiveness of the framework largely depends on the availability and quality of metadata, as demonstrated by the performance improvements when metadata is included. Does this dependency restrict the model's applicability in scenarios where metadata is unavailable or of poor quality? Could it be worthwhile to consider adding a comparison of the model's performance using only text to address this dependency?

**Suitability:**

3

---

### Official Review · Reviewer_wAA7 · 2024-05-24

**Rating:** 4
**Confidence:** 3

**Summary:**

The paper proposes a framework for soundscape mapping by utilizing multi-scale satellite imagery, audio, and text to create joint representations. The framework aims to capture the uncertainty in the soundscape by designing the probabilistic embedding space. It also integrates metadata like geolocation, time, and data sources to enable learning dynamic representations of soundscapes. The paper introduces the GeoSound dataset, which consists of over 300k geotagged audio samples paired with satellite imagery.

**Strengths:**

[+] The motivation for metadata-aware and probabilistic embedding space is intuitive and meaningful.
[+] The proposed GeoSound dataset contributes significantly to the development of the field.
[+] Experimental results confirm the effectiveness of motivation and the proposed method.

**Limitations:**

[-] I found that the evaluation metric R@10% in this paper is inconsistent with the R@100 in reference [1], and the same evaluation metric was not employed.

[-] The existing paper [2] proposed probabilistic embeddings in 2021, thereby weakening the novelty of the probabilistic embedding space.

[1]Subash Khanal, Srikumar Sastry, Aayush Dhakal, and Nathan Jacobs. 2023. Learn
ing Tri-modal Embeddings for Zero-Shot Soundscape Mapping. In British Machine Vision Conference (BMVC)

[2]Sanghyuk Chun, Seong Joon Oh, Rafael Sampaio De Rezende, Yannis Kalantidis, and Diane Larlus. 2021. Probabilistic embeddings for cross-modal retrieval. In Proceedings of the IEEE/CVF Conference on Computer Vision and Pattern Recognition.

**Suitability:**

3

---

### Official Review · Reviewer_bKvE · 2024-05-28

**Rating:** 3
**Confidence:** 1

**Summary:**

This work presents a new method that learns probabilistic embeddings for zero-shot soundscape mapping.

**Strengths:**

- A large-scale dataset is introduced to facilitate the research in the field.
- SOTA results are achieved on several datasets.

**Limitations:**

- Zero-shot learning normally involves splitting data into seen and unseen classes. Does this task have such settings?
- The motivations behind the design of Equation (7) need to be included. Why does it reduce the distance between \mu while making \sigma approach zero?
- The comparisons between the proposed PSM and baseline should be reflected in Figures 2-4.

Minus suggestions:
- “PSM: Probabilistic” is out of size.
- Figure 1 is unclear. What is the prediction or output of the framework?

**Suitability:**

3

---

### Meta-Review · Area_Chair_TMvv · 2024-07-03

**Recommendation:** Accept (Poster)
**Confidence:** 5

**Metareview:**

The paper studies soundscape mapping, which aims to generate a distribution of sounds for any location on the Earth. The paper contributes a new large dataset and a baseline that combines text, image, and audio. The paper received 2 BAs and 1 WA. The reviewers liked the good motivation for the method, the usefulness of a large dataset, and convincing experiments. The AC agrees with the reviewers and thinks the problem is quite interesting and the contribution is solid.